# Tracking the Best Expert in Non-stationary Stochastic Environments

**Chen-Yu Wei**     **Yi-Te Hong**     **Chi-Jen Lu**
Institute of Information Science
Academia Sinica, Taiwan
{bahh723, ted0504, cjlu}@iis.sinica.edu.tw

## Abstract

We study the dynamic regret of multi-armed bandit and experts problem in non-stationary stochastic environments. We introduce a new parameter $\Lambda$, which measures the total statistical variance of the loss distributions over $T$ rounds of the process, and study how this amount affects the regret. We investigate the interaction between $\Lambda$ and $\Gamma$, which counts the number of times the distributions change, as well as $\Lambda$ and $V$, which measures how far the distributions deviates over time. One striking result we find is that even when $\Gamma$, $V$, and $\Lambda$ are all restricted to constant, the regret lower bound in the bandit setting still grows with $T$. The other highlight is that in the full-information setting, a constant regret becomes achievable with constant $\Gamma$ and $\Lambda$, as it can be made independent of $T$, while with constant $V$ and $\Lambda$, the regret still has a $T^{1/3}$ dependency. We not only propose algorithms with upper bound guarantee, but prove their matching lower bounds as well.

## 1   Introduction

Many situations in our daily life require us to make repeated decisions which result in some losses corresponding to our chosen actions. This can be abstracted as the well-known online decision problem in machine learning [5]. Depending on how the loss vectors are generated, two different worlds are usually considered. In the adversarial world, loss vectors are assumed to be deterministic and controlled by an adversary, while in the stochastic world, loss vectors are assumed to be sampled independently from some distributions. In both worlds, good online algorithms are known which can achieve a regret of about $\sqrt{T}$ over $T$ time steps, where the regret is the difference between the total loss of the online algorithm and that of the best offline one. Another distinction is about the information the online algorithm can receive after each action. In the full-information setting, it gets to know the whole loss vector of that step, while in the bandit setting, only the loss value of the chosen action is received. Again, in both settings, a regret of about $\sqrt{T}$ turns out to be achievable.

While the regret bounds remain in the same order in those general scenarios discussed above, things become different when some natural conditions are considered. One well-known example is that in the stochastic multi-armed bandit (MAB) problem, when the best arm (or action) is substantially better than the second best, with a constant gap between their means, then a much lower regret, of the order of $\log T$, becomes possible. This motivates us to consider other possible conditions which can have finer characterization of the problem in terms of the achievable regret.

In the stochastic world, most previous works focused on the stationary setting, in which the loss (or reward) vectors are assumed to be sampled from the same distribution for all time steps. With this assumption, although one needs to balance between exploration and exploitation in the beginning, after some trials, one can be confident about which action is the best and rest assured that there are no more surprises. On the other hand, the world around us may not be stationary, in which existing learning algorithms for the stationary case may no longer work. In fact, in a non-stationary world, the

dilemma between exploration and exploitation persists as the underlying distribution may drift as time evolves. How does the non-stationarity affect the achievable regret? How does one measure the degree of non-stationarity?

In this paper, we answer the above questions through the notion of dynamic regret, which measures the algorithm's performance against an offline algorithm allowed to select the best arm at every step.

**Related Works.** One way to measure the non-stationarity of a sequence of distributions is to count the number of times the distribution at a time step differs from its previous one. Let $\Gamma - 1$ be this number so that the whole time horizon can be partitioned into $\Gamma$ intervals, with each interval having a stationary distribution. In the bandit setting, a regret of about $\sqrt{\Gamma T}$ is achieved by the EXP3.S algorithm in [2], as well as the discounted UCB and sliding-window UCB algorithms in [8]. The dependency on $T$ can be refined in the full-information setting: AdaNormalHedge [10] and Adapt-ML-Prod [7] can both achieve regret in the form of $\sqrt{\Gamma C}$, where $C$ is the total first-order and second-order excess loss respectively, which is upper-bounded by $T$. From a slightly different Online Mirror Descent approach, [9] can also achieve a regret of about $\sqrt{\Gamma D}$, where $D$ is the sum of differences between consecutive loss vectors.

Another measure of non-stationarity, denoted by $V$, is to compute the difference between the means of consecutive distributions and sum them up. Note that this allows the possibility for the best arm to change frequently, with a very large $\Gamma$, while still having similar distributions with a small $V$. For such a measure $V$, [3] provided a bandit algorithm which achieves a regret of about $V^{1/3}T^{2/3}$. This regret upper bound is unimprovable in general even in the full-information setting, as a matching lower bound was shown in [4]. Again, [9] refined the upper bound in the full-information setting through the introduction of $D$, achieving the regret of about $\sqrt[3]{\tilde{V}DT}$, for a parameter $\tilde{V}$ different but related to $V$: $\tilde{V}$ calculates the sum of differences between consecutive *realized* loss vectors, while $V$ measures that between *mean* loss vectors. This makes the results of [3] and [9] incomparable. The problem stems from the fact that [9] considers the traditional adversarial setting, while [3] studies the non-stationary stochastic setting. In this paper, we will provide a framework that bridges these two seemingly disparate worlds.

**Our Results.** We base ourselves in the stochastic world with non-stationary distributions, characterized by the parameters $\Gamma$ and $V$. In addition, we introduce a new parameter $\Lambda$, which measures the total statistical variance of the distributions. Note that traditional adversarial setting corresponds to the case with $\Lambda = 0$ and $\Gamma \approx V \approx T$, while the traditional stochastic setting has $\Lambda \approx T$ and $\Gamma = V = 1$. Clearly, with a smaller $\Lambda$, the learning problem becomes easier, and we would like to understand the tradeoff between $\Lambda$ and other parameters, including $\Gamma$, $V$, and $T$. In particular, we would like to know how the bounds described in the related works would be changed. Would all the dependency on $T$ be replaced by $\Lambda$, or would only some partial dependency on $T$ be shifted to $\Lambda$?

First, we consider the effect of the variance $\Lambda$ with respect to the parameter $\Gamma$. We show that in the full-information setting, a regret of about $\sqrt{\Gamma\Lambda} + \Gamma$ can be achieved, which is independent of $T$. On the other hand, we show a sharp contrast that in the bandit setting, the dependency on $T$ is unavoidable, and a lower bound of the order of $\sqrt{\Gamma T}$ exists. That is, even when there is no variance in distributions, with $\Lambda = 0$, and the distributions only change once, with $\Gamma = 2$, any bandit algorithm cannot avoid a regret of about $\sqrt{T}$, while a full-information algorithm can achieve a constant regret independent of $T$.

Next, we study the tradeoff between $\Lambda$ and $V$. We show that in the bandit setting, a regret of about $\sqrt[3]{\Lambda VT} + \sqrt{VT}$ is achievable. Note that this recovers the $V^{1/3}T^{2/3}$ regret bound of [3] as $\Lambda$ is at most of the order of $T$, but our bound becomes better when $\Lambda$ is much smaller than $T$. Again, one may notice the dependency on $T$ and wonder if this can also be removed in the full-information setting. We show that in the full-information setting, the regret upper bound and lower bound are both about $\sqrt[3]{\Lambda VT} + V$. Our upper bound is incomparable to the $\sqrt[3]{\tilde{V}DT}$ bound of [9], since their adversarial setting corresponds to $\Lambda = 0$ and their $D$ can be as large as $T$ in our setting. Moreover, we see that while the full-information regret bound is slightly better than that in the bandit setting, there is still an unavoidable $T^{1/3}$ dependency.

Our results provide a big picture of the regret landscape in terms of the parameters $\Lambda, \Gamma, V$, and $T$, in both full-information and bandit settings. A table summarizing our bounds as well as previous ones is given in Appendix A in the supplementary material. Finally, let us remark that our effort mostly focuses on characterizing the achievable (minimax) regrets, and most of our upper bounds are achieved by algorithms which need the knowledge of the related parameters and may not be practical. To complement this, we also propose a parameter-free algorithm, which still achieve a good regret bound and may have independent interest of its own.

## 2 Preliminaries

Let us first introduce some notations. For an integer $K > 0$, let $[K]$ denote the set $\{1, \ldots, K\}$. For a vector $\ell \in \mathbb{R}^K$, let $\ell_i$ denote its $i$'th component. When we need to refer to a time-indexed vector $\ell_t \in \mathbb{R}^K$, we will write $\ell_{t,i}$ to denote its $i$'th component. We will use the indicator function $\mathbb{1}_{\mathcal{C}}$ for a condition $\mathcal{C}$, which gives the value 1 if $\mathcal{C}$ holds and 0 otherwise. For a vector $\ell$, we let $\|\ell\|_b$ denote its $L_b$-norm. While standard notation $\mathcal{O}(\cdot)$ is used to hide constant factors, we will use the notation $\tilde{\mathcal{O}}(\cdot)$ to hide logarithmic factors.

Next, let us describe the problem we study in this paper. Imagine that a learner is given the choice of a total of $K$ actions, and has to play iteratively for a total of $T$ steps. At step $t$, the learner needs to choose an action $a_t \in [K]$, and then suffers a corresponding loss $\ell_{t,i} \in [0, 1]$, which is independently drawn from a non-stationary distribution with expected loss $\mathbb{E}[\ell_{t,i}] = \mu_{t,i}$, which may drift over time. After that, the learner receives some feedback from the environment. In the full-information setting, the feedback gives the whole loss vector $\ell_t = (\ell_{t,1}, ..., \ell_{t,K})$, while in the bandit setting, only the loss $\ell_{t,a_t}$ of the chosen action is revealed. A standard way to evaluate the learner's performance is to measure her (or his) regret, which is the difference between the total loss she suffers and that of an offline algorithm. While most prior works consider offline algorithms which can only play a fixed action for all the steps, we consider stronger offline algorithms which can take different actions in different steps. Our consideration is natural for non-stationary distributions, although this would make the regret large when compared to such stronger offline algorithms. Formally, we measure the learner's performance by its expected *dynamic pseudo-regret*, defined as $\sum_{t=1}^{T} \mathbb{E}\left[\ell_{t,a_t} - \ell_{t,u_t^*}\right] = \sum_{t=1}^{T} \left(\mu_{t,a_t} - \mu_{t,u_t^*}\right)$, where $u_t^* = \arg\min_i \mu_{t,i}$ is the best action at step $t$. For convenience, we will simply refer it as the regret of the learner later in our paper.

We will consider the following parameters characterizing different aspects of the environments:

$$\Gamma = 1 + \sum_{t=2}^{T} \mathbb{1}_{\mu_t \neq \mu_{t-1}}, V = \sum_{t=1}^{T} \|\mu_t - \mu_{t-1}\|_\infty, \text{ and } \Lambda = \sum_{t=1}^{T} \mathbb{E}\left[\|\ell_t - \mu_t\|_2^2\right], \tag{1}$$

where we let $\mu_0$ be the all-zero vector. Here, $\Gamma - 1$ is the number of times the distributions switch, $V$ measures the distance the distributions deviate, and $\Lambda$ is the total statistical variance of these $T$ distributions. We will call distributions with a small $\Gamma$ switching distributions, while we will call distributions with a small $V$ drifting distributions and call $V$ the total drift of the distributions.

Finally, we will need the following large deviation bound, known as empirical Bernstein inequality.

**Theorem 2.1.** *[11] Let $X = (X_1, ..., X_n)$ be a vector of independent random variables taking values in $[0, 1]$, and let $\Lambda_X = \sum_{1 \leq i < j \leq n} (X_i - X_j)^2 / (n(n-1))$. Then for any $\delta > 0$, we have*

$$\Pr\left[\left|\sum_{i=1}^{n} \frac{\mathbb{E}[X_i] - X_i}{n}\right| > \rho(n, \Lambda_X, \delta)\right] \leq \delta, \text{ for } \rho(n, \Lambda, \delta) = \sqrt{\frac{2\Lambda \log \frac{2}{\delta}}{n}} + \frac{7 \log \frac{2}{\delta}}{3(n-1)}.$$

## 3 Algorithms

We would like to characterize the achievable regret bounds for both switching and drifting distributions, in both full-information and bandit settings. In particular, we would like to understand the interplay among the parameters $\Gamma, V, \Lambda$, and $T$, defined in (1). The only known upper bound which is good enough for our purpose is that by [8] for switching distributions in the bandit setting, which is close to the lower bound in our Theorem 4.1. In subsection 3.1, we provide a bandit algorithm for drifting distributions which achieves an almost optimal regret upper bound, when given the parameters

---

**Algorithm 1** Rerun-UCB-V

---

**Initialization:** Set $B$ according to (2) and $\delta = 1/(KT)$.
**for** $m = 1, \ldots, T/B$ **do**
    **for** $t = (m-1)B + 1, \ldots, mB$ **do**
        Choose arm $a_t := \operatorname{argmin}_i(\hat{\mu}_{t,i} - \lambda_{t,i})$, with $\hat{\mu}_{t,i}$ and $\lambda_{t,i}$ computed according to (3).
    **end for**
**end for**

---

$V, \Lambda, T$. In subsection 3.2, we provide a full-information algorithm which works for both switching and drifting distributions. The regret bounds it achieves are also close to optimal, but it again needs the knowledge of the related parameters. To complement this, we provide a full-information algorithm in subsection 3.3, which does not need to know the parameters but achieves slightly larger regret bounds.

### 3.1 Parameter-Dependent Bandit Algorithm

In this subsection, we consider drifting distributions parameterized by $V$ and $\Lambda$. Our main result is a bandit algorithm which achieves a regret of about $\sqrt[3]{\Lambda V T} + \sqrt{V T}$. As we aim to achieve smaller regrets for distributions with smaller statistical variances, we adopt a variant of the UCB algorithm developed by [1], called UCB-V, which takes variances into account when building its confidence interval.

Our algorithm divides the time steps into $T/B$ intervals $\mathcal{I}_1, \ldots, \mathcal{I}_{T/B}$, each having $B$ steps,[1] with

$$B = \sqrt[3]{K^2 \Lambda T / V^2} \text{ if } K\Lambda^2 \geq TV \text{ and } B = \sqrt{KT/V} \text{ otherwise.} \qquad (2)$$

For each interval, our algorithm clears all the information from previous intervals, and starts a fresh run of UCB-V. More precisely, before step $t$ in an interval $\mathcal{I}$, it maintains for each arm $i$ its empirical mean $\hat{\mu}_{t,i}$, empirical variance $\hat{\Lambda}_{t,i}$, and size of confidence interval $\lambda_{t,i}$, defined as

$$\hat{\mu}_{t,i} = \sum_{s \in S_{t,i}} \frac{\ell_{s,i}}{|S_{t,i}|}, \; \hat{\Lambda}_{t,i} = \sum_{r,s \in S_{t,i}} \frac{(\ell_{r,i} - \ell_{s,i})^2}{|S_{t,i}|(|S_{t,i}| - 1)}, \; \text{ and } \lambda_{t,i} = \rho(|S_{t,i}|, \hat{\Lambda}_{t,i}, \delta), \qquad (3)$$

where $S_{t,i}$ denotes the set of steps before $t$ in $\mathcal{I}$ that arm $i$ was played, and $\rho$ is the function given in Theorem 2.1. Here we use the convention that $\hat{\mu}_{t,i} = 0$ if $|S_{t,i}| = 0$, while $\hat{\Lambda}_{t,i} = 0$ and $\lambda_{t,i} = 1$ if $|S_{t,i}| \leq 1$. Then at step $t$, our algorithm selects the optimistic arm

$$a_t := \operatorname*{argmin}_i(\hat{\mu}_{t,i} - \lambda_{t,i}),$$

receives the corresponding loss, and updates the statistics.

Our algorithm is summarized in Algorithm 1, and its regret is guaranteed by the following, which we prove in Appendix B in the supplementary material.

**Theorem 3.1.** *The expected regret of Algorithm 1 is at most $\tilde{\mathcal{O}}(\sqrt[3]{K^2 \Lambda V T} + \sqrt{KVT})$.*

### 3.2 Parameter-Dependent Full-Information Algorithms

In this subsection, we provide full-information algorithms for switching and drifting distributions. In fact, they are based on an existing algorithm from [6], which is known to work in a different setting: the loss vectors are deterministic and adversarial, and the offline comparator cannot switch arms. In that setting, one of their algorithms, based on gradient-descent (GD), can achieve a regret of $\mathcal{O}(\sqrt{D})$ where $D = \sum_t \|\ell_t - \ell_{t-1}\|_2^2$, which is small when the loss vectors have small deviation. Our first observation is that their algorithm in fact can work against a dynamic offline comparator which switches arms less than $N$ times, given any $N$, with its regret becoming $\mathcal{O}(\sqrt{ND})$. Our second observation is that when $\Lambda$ is small, each observed loss vector $\ell_t$ is likely to be close to its true mean

**Algorithm 2** Full-information GD-based algorithm

---

**Initialization:** Let $x_1 = \hat{x}_1 = (1/K, \ldots, 1/K)^\top$.
**for** $t = 1, 2, \ldots, T$ **do**
    Play $\hat{x}_t = \arg\min_{\hat{x} \in \mathcal{X}}(\langle \ell_{t-1}, \hat{x} \rangle + \frac{1}{\eta_t}\|\hat{x} - x_t\|_2^2)$, and then receive loss vector $\ell_t$.
    Update $x_{t+1} = \arg\min_{x \in \mathcal{X}}(\langle \ell_t, x \rangle + \frac{1}{\eta_t}\|x - x_t\|_2^2)$.
**end for**

---

$\mu_t$, and when $V$ is small, $\ell_t$ is likely to be close to $\ell_{t-1}$. These two observations make possible for us to adopt their algorithm to our setting.

We show the first algorithm in Algorithm 2, with the feasible set $\mathcal{X}$ being the probability simplex. The idea is to use $\ell_{t-1}$ as an estimate for $\ell_t$ to move $\hat{x}_t$ further in a possibly beneficial direction. Its regret is guaranteed by the following, which we prove in Appendix C in the supplementary material.

**Theorem 3.2.** *For switching distributions parameterized by $\Gamma$ and $\Lambda$, the regret of Algorithm 2 with $\eta_t = \eta = \sqrt{\Gamma/(\Lambda + K\Gamma)}$, is at most $\mathcal{O}(\sqrt{\Gamma\Lambda} + \sqrt{K\Gamma})$.*

Note that for switching distributions, the regret of Algorithm 2 does not depend on $T$, which means that it can achieve a constant regret for constant $\Gamma$ and $\Lambda$. Let us remark that although using a variant based on multiplicative updates could result in a better dependency on $K$, an additional factor of $\log T$ would then emerge when using existing techniques for dealing with dynamic comparators.

For drifting distributions, one can show that Algorithm 2 still works and has a good regret bound. However, a slightly better bound can be achieved as we describe next. The idea is to divide the time steps into $T/B$ intervals of size $B$, with $B = \sqrt[3]{\Lambda T/V^2}$ if $\Lambda T > V^2$ and $B = 1$ otherwise, and re-run Algorithm 2 in each interval with an adaptive learning rate. One way to have an adaptive learning rate can be found in [9], which works well when there is only one interval. A natural way to adopt it here is to reset the learning rate at the start of each interval, but this does not lead to a good enough regret bound as it results in some constant regret at the start of every interval. To avoid this, some careful changes are needed. Specifically, in an interval $[t_1, t_2]$, we run Algorithm 2 with the learning rate reset as

$$\eta_t = 1/\sqrt{4 \sum_{\tau=t_1}^{t-1} \|\ell_\tau - \ell_{\tau-1}\|_2^2}$$

for $t > t_1$, with $\eta_{t_1} = \infty$ initially for every interval. This has the benefit of having small or even no regret at the start of an interval when the loss vectors across the boundary have small or no deviation. The regret of this new algorithm is guaranteed by the following, which we prove in Appendix D in the supplementary material.

**Theorem 3.3.** *For drifting distributions parameterized by $V$ and $\Lambda$, the regret of this new algorithm is at most $\mathcal{O}(\sqrt[3]{V\Lambda T} + \sqrt{KV})$.*

### 3.3 Parameter-Free Full-Information Algorithm

The reason that our algorithm for Theorem 3.3 needs the related parameters is to set its learning rate properly. To have a parameter-free algorithm, we would like to adjust the learning rate dynamically in a data-driven way. One way for doing this can be found in [7], which is based on the multiplicative updates variant of the mirror-descent algorithm. It achieves a static regret of about $\sqrt{\sum_t r_{t,k}^2}$ against any expert $k$, where $r_{t,k} = \langle p_t, \ell_t \rangle - \ell_{t,k}$ is its instantaneous regret for playing $p_t$ at step $t$. However, in order to work in our setting, we would like the regret bound to depend on $\ell_t - \ell_{t-1}$ as seen previously. This suggests us to modify the Adapt-ML-Prod algorithm of [7] using the idea of [6], which takes $\ell_{t-1}$ as an estimate of $\ell_t$ to move $p_t$ further in an optimistic direction.

Recall that the algorithm of [7] maintains a separate learning rate $\eta_{t,k}$ for each arm $k$ at time $t$, and it updates the weight $w_{t,k}$ as well as $\eta_{t,k}$ using the instantaneous regret $r_{t,k}$. To modify the algorithm using the idea of [6], we would like to have an estimate $m_{t,k}$ for $r_{t,k}$ in order to move $p_{t,k}$ further using $m_{t,k}$ and update the learning rate accordingly. More precisely, at step $t$, we now play $p_t$, with

$$p_{t,k} = \eta_{t-1,k}\tilde{w}_{t-1,k}/\langle \eta_{t-1}, \tilde{w}_{t-1} \rangle \quad \text{where } \tilde{w}_{t-1,k} = w_{t-1,k}\exp(\eta_{t-1,k}m_{t,k}), \tag{4}$$

---

**Algorithm 3** Optimistic-Adapt-ML-Prod

---
**Initialization:** Let $w_{0,k} = 1/K$ and $\ell_{0,k} = 0$ for every $k \in [K]$.
**for** $t = 1, 2, \ldots, T$ **do**
    Play $p_t$ according to (4), and then receive loss vector $\ell_t$.
    Update each weight $w_{t,k}$ according to (5) and each learning rate $\eta_{t,k}$ according to (6).
**end for**

---

which uses the estimate $m_{t,k}$ to move further from $w_{t-1,k}$. Then after receiving the loss vector $\ell_t$, we update each weight

$$w_{t,k} = \left( w_{t-1,k} \exp \left( \eta_{t-1,k} r_{t,k} - \eta_{t-1,k}^2 (r_{t,k} - m_{t,k})^2 \right) \right)^{\eta_{t,k}/\eta_{t-1,k}} \tag{5}$$

as well as each learning rate

$$\eta_{t,k} = \min \left\{ 1/4, \sqrt{(\ln K) / \left( 1 + \sum_{s \in [t]} (r_{s,k} - m_{s,k})^2 \right)} \right\}. \tag{6}$$

Our algorithm is summarized in Algorithm 3, and we will show that it achieves a regret of about $\sqrt{\sum_t (r_{t,k} - m_{t,k})^2}$ against arm $k$. It remains to choose an appropriate estimate $m_{t,k}$. One attempt is to have $m_{t,k} = r_{t-1,k}$, but $r_{t,k} - r_{t-1,k} = (\langle p_t, \ell_t \rangle - \ell_{t,k}) - (\langle p_{t-1}, \ell_{t-1} \rangle - \ell_{t-1,k})$, which does not lead to a desirable bound. The other possibility is to set $m_{t,k} = \langle p_t, \ell_{t-1} \rangle - \ell_{t-1,k}$, which can be shown to have $(r_{t,k} - m_{t,k})^2 \leq (2\|\ell_t - \ell_{t-1}\|_\infty)^2$. However, it is not clear how to compute such $m_{t,k}$ because it depends on $p_{t,k}$ which in turns depends on $m_{t,k}$ itself. Fortunately, we can approximate it efficiently in the following way.

Note that the key quantity is $\langle p_t, \ell_{t-1} \rangle$. Given its value $\alpha$, $\tilde{w}_{t-1,k}$ and $p_{t,k}$ can be seen as functions of $\alpha$, defined according to (5) as $\tilde{w}_{t-1,k}(\alpha) = w_{t-1,k} \exp(\eta_{t,k}(\alpha - \ell_{t-1,k}))$ and $p_{t,k}(\alpha) = \eta_{t-1,k} \tilde{w}_{t-1,k}(\alpha) / \sum_i \eta_{t-1,i} \tilde{w}_{t-1,i}(\alpha)$. Then we would like to show the existence of $\alpha$ such that $\langle p_t(\alpha), \ell_{t-1} \rangle = \alpha$ and to find it efficiently. For this, consider the function $f(\alpha) = \langle p_t(\alpha), \ell_{t-1} \rangle$, with $p_t(\alpha)$ defined above. It is easy to check that $f$ is a continuous function bounded in $[0, 1]$, which implies the existence of some fixed point $\alpha \in [0, 1]$ with $f(\alpha) = \alpha$. Using a binary search, such an $\alpha$ can be approximated within error $1/T$ in $\log T$ iterations. As such a small error does not affect the order of the regret, we will ignore it for simplicity of presentation, and assume that we indeed have $\langle p_t, \ell_{t-1} \rangle$ and hence $m_{t,k} = \langle p_t, \ell_{t-1} \rangle - \ell_{t-1,k}$ without error.

Then we have the following regret bound (c.f. [7, Corollary 4]), which we prove in Appendix E in the supplementary material.

**Theorem 3.4.** *The static regret of Algorithm 3 w.r.t. any arm (or expert) $k \in [K]$ is at most*

$$\hat{\mathcal{O}} \left( \sqrt{\sum_{t \in [T]} (r_{t,k} - m_{t,k})^2 \ln K} + \ln K \right) \leq \hat{\mathcal{O}} \left( \sqrt{\sum_{t \in [T]} \|\ell_t - \ell_{t-1}\|_\infty^2 \ln K} + \ln K \right),$$

*where the notation $\hat{\mathcal{O}}(\cdot)$ hides a $\ln \ln T$ factor.*

The regret in the theorem above is measured against a fixed arm. To achieve a dynamic regret against an offline algorithm which can switch arms, one can use a generic reduction to the so-called sleeping experts problem. In particular, we can use the idea in [7] by creating $\tilde{K} = KT$ sleeping experts, and run our Algorithm 3 on these $\tilde{K}$ experts (instead of on the $K$ arms). More precisely, each sleeping expert is indexed by some pair $(s, k)$, and it is asleep for steps before $s$ and becomes awake for steps $t \geq s$. At step $t$, it calls Algorithm 3 for the distribution $\tilde{p}_t$ over the $\tilde{K}$ experts, and computes its own distribution $p_t$ over $K$ arms, with $p_{t,k}$ proportional to $\sum_{s=1}^t \tilde{p}_{t,(s,k)}$. Then it plays $p_t$, receives loss vector $\ell_t$, and feeds some modified loss vector $\tilde{\ell}_t$ and estimate vector $\tilde{m}_t$ to Algorithm 3 for update. Here, we set $\tilde{\ell}_{t,(s,k)}$ to its expected loss $\langle p_t, \ell_t \rangle$ if expert $(s, k)$ is asleep and to $\ell_{t,k}$ otherwise, while we set $\tilde{m}_{t,(s,k)}$ to 0 if expert $(s, k)$ is asleep and to $m_{t,k} = \langle p_t, \ell_{t-1} \rangle - \ell_{t-1,k}$ otherwise. This choice allows us to relate the regret of Algorithm 3 to that of the new algorithm, which can be seen in the proof of the following theorem, given in Appendix F in the supplementary material.

**Theorem 3.5.** *The dynamic expected regret of the new algorithm is $\tilde{\mathcal{O}}(\sqrt{\Gamma \Lambda \ln K} + \Gamma \ln K)$ for switching distributions and $\tilde{\mathcal{O}}(\sqrt[3]{V \Lambda T \ln K} + \sqrt{V T \ln K})$ for drifting distributions.*

## 4 Lower Bounds

We study regret lower bounds in this section. In subsection 4.1, we show that for switching distributions with $\Gamma - 1 \geq 1$ switches, there is an $\Omega(\sqrt{\Gamma T})$ lower bound for bandit algorithms, even when there is no variance ($\Lambda = 0$) and there are constant loss gaps between the optimal and suboptimal arms. We also show a full-information lower bound, which almost matches our upper bound in Theorem 3.2. In subsection 4.2, we show that for drifting distributions, our upper bounds in Theorem 3.1 and Theorem 3.2 are almost tight. In particular, we show that now even for full-information algorithms, a large $\sqrt[3]{T}$ dependency in the regret turns out to be unavoidable, even for small $V$ and $\Lambda$. This provides a sharp contrast to the upper bound of our Theorem 3.2, which shows that a constant regret is in fact achievable by a full-information algorithm for switching distributions with constant $\Gamma$ and $\Lambda$. For simplicity of presentation, we will only discuss the case with $K = 2$ actions, as it is not hard to extend our proofs to the general case.

### 4.1 Switching Distributions

In contrast to the full-information setting, the existence of switches presents a dilemma with lose-lose situation for a bandit algorithm: in order to detect any possible switch early enough, it must explore aggressively, but this has the consequence of playing suboptimal arms too often. To fool any bandit algorithm, we will switch between two deterministic distributions, with no variance, which have mean vectors $\ell^{(1)} = (1/2, 1)^\top$ and $\ell^{(2)} = (1/2, 0)^\top$, respectively. Our result is the following.

**Theorem 4.1.** *The worst-case expected regret of any bandit algorithm is* $\Omega(\sqrt{\Gamma T})$, *for* $\Gamma \geq 2$.

*Proof.* Consider any bandit algorithm $\mathcal{A}$, and let us partition the $T$ steps into $\Gamma/2$ intervals, each consisting of $B = 2T/\Gamma$ steps. Our goal is to make $\mathcal{A}$ suffer in each interval an expected regret of $\Omega(\sqrt{B})$ by switching the loss vectors at most once. As mentioned before, we will only switch between two different deterministic distributions with mean vectors: $\ell^{(1)}$ and $\ell^{(2)}$. Note that we can see these two distributions simply as two loss vectors, with $\ell^{(i)}$ having arm $i$ as the optimal arm.

In what follows, we focus on one of the intervals, and assume that we have chosen the distributions in all previous intervals. We would like to start the interval with the loss vector $\ell^{(1)}$. Let $N_2$ denote the expected number of steps $\mathcal{A}$ plays the suboptimal arm 2 in this interval if $\ell^{(1)}$ is used for the whole interval. If $N_2 \geq \sqrt{B}/2$, we can actually use $\ell^{(1)}$ for the whole interval with no switch, which makes $\mathcal{A}$ suffer an expected regret of at least $(1/2) \cdot \sqrt{B}/2 = \sqrt{B}/4$ in this interval. Thus, it remains to consider the case with $N_2 < \sqrt{B}/2$. In this case, $\mathcal{A}$ does not explore arm 2 often enough, and we let it pay by choosing an appropriate step to switch to the other loss vector $\ell^{(2)} = (1/2, 0)^\top$, which has arm 2 as the optimal one. For this, let us divide the $B$ steps of the interval into $\sqrt{B}$ blocks, each consisting of $\sqrt{B}$ steps. As $N_2 < \sqrt{B}/2$, there must be a block in which the expected number of steps that $\mathcal{A}$ plays arm 2 is at most $N_2/\sqrt{B} < 1/2$. By a Markov inequality, the probability that $\mathcal{A}$ ever plays arm 2 in this block is less than $1/2$. This implies that when given the loss vector $\ell^{(1)}$ for all the steps till the end of this block, $\mathcal{A}$ never plays arm 2 in the block with probability more than $1/2$. Therefore, if we make the switch to the loss vector $\ell^{(2)} = (1/2, 0)^\top$ at the beginning of the block, then $\mathcal{A}$ with probability more than $1/2$ still never plays arm 2 and never notices the switch in this block. As arm 2 is the optimal one with respect to $\ell^{(2)}$, the expected regret of $\mathcal{A}$ in this block is more than $(1/2) \cdot (1/2) \cdot \sqrt{B} = \sqrt{B}/4$.

Now if we choose distributions in each interval as described above, then there are at most $\Gamma/2 \cdot 2 = \Gamma$ periods of stationary distribution in the whole horizon, and the total expected regret of $\mathcal{A}$ can be made at least $\Gamma/2 \cdot \sqrt{B}/4 = \Gamma/2 \cdot \sqrt{2T/\Gamma}/4 = \Omega(\sqrt{\Gamma T})$, which proves the theorem. $\qquad\qquad\square$

For full-information algorithms, we have the following lower bound, which almost matches our upper bound in Theorem 3.2. We provide the proof in Appendix G in the supplementary material.

**Theorem 4.2.** *The worst-case expected regret of any full-information algorithm is* $\Omega(\sqrt{\Gamma\Lambda} + \Gamma)$.

## 4.2 Drifting Distributions

In this subsection, we show that the regret upper bounds achieved by our bandit algorithm and full-information algorithm are close to optimal by showing almost matching lower bounds. More precisely, we have the following.

**Theorem 4.3.** *The worst-case expected regret of any full-information algorithm is $\Omega(\sqrt[3]{\Lambda V T} + V)$, while that of any bandit algorithm is $\Omega(\sqrt[3]{\Lambda V T} + \sqrt{VT})$.*

*Proof.* Let us first consider the full-information case. When $\Lambda T \leq 32KV^2$, we immediately have from Theorem 4.2 the regret lower bound of $\Omega(\Gamma) \geq \Omega(V) \geq \Omega(\sqrt[3]{\Lambda V T} + V)$.

Thus, let us focus on the case with $\Lambda T \geq 32KV^2$. In this case, $V \leq \mathcal{O}(\sqrt[3]{\Lambda V T})$, so it suffices to prove a lower bound of $\Omega(\sqrt[3]{\Lambda V T})$. Fix any full-information algorithm $\mathcal{A}$, and we will show the existence of a sequence of loss distributions for $\mathcal{A}$ to suffer such an expected regret. Following [3], we divide the time steps into $T/B$ intervals of length $B$, and we set $B = \sqrt[3]{\Lambda T/(32KV^2)} \geq 1$. For each interval, we will pick some arm $i$ as the optimal one, and give it some loss distribution $\mathcal{P}$, while other arms are sub-optimal and all have some loss distribution $\mathcal{Q}$. We need $\mathcal{P}$ and $\mathcal{Q}$ to satisfy the following three conditions: (a) $\mathcal{P}$'s mean is smaller than $\mathcal{Q}$'s by $\epsilon$, (b) their variances are at most $\sigma^2$, and (c) their KL divergence satisfies $(\ln 2)\mathrm{KL}(\mathcal{Q},\mathcal{P}) \leq \epsilon^2/\sigma^2$, for some $\epsilon, \sigma \in (0, 1)$ to be specified later. Their existence is guaranteed by the following, which we prove in Appendix H in the supplementary material.

**Lemma 4.4.** *For any $0 \leq \sigma \leq 1/2$ and $0 \leq \epsilon \leq \sigma/\sqrt{2}$, there exist distributions $\mathcal{P}$ and $\mathcal{Q}$ satisfying the three conditions above.*

Let $\mathcal{D}_i$ denote the joint distribution of such $K$ distributions, with arm $i$ being the optimal one, and we will use the same $\mathcal{D}_i$ for all the steps in an interval. We will show that for any interval, there is some $i$ such that using $\mathcal{D}_i$ this way can make algorithm $\mathcal{A}$ suffer a large expected regret in the interval, conditioned on the distributions chosen for previous intervals. Before showing that, note that when we choose distributions in this way, their total variance is at most $TK\sigma^2$ while their total drift is at most $(T/B)\epsilon$. To have them bounded by $\Lambda$ and $V$ respectively, we choose $\sigma = \sqrt{\Lambda/(4KT)}$ and $\epsilon = VB/T$, which satisfy the condition of Lemma 4.4, with our choice of $B$.

To find the distributions, we deal with the intervals one by one. Consider any interval, and assume that the distributions for previous intervals have been chosen. Let $N_i$ denote the number of steps $\mathcal{A}$ plays arm $i$ in this interval, and let $\mathbb{E}_i[N_i]$ denote its expectation when $\mathcal{D}_i$ is used for every step of the interval, conditioned on the distributions of previous intervals. One can bound this conditional expectation in terms of a related one, denoted as $\mathbb{E}_{unif}[N_i]$, when every arm has the distribution $\mathcal{Q}$ for every step of the interval, again conditioned on the distributions of previous intervals. Specifically, using an almost identical argument to that in [2, proof of Theorem A.2.], one can show that

$$\mathbb{E}_i[N_i] \leq \mathbb{E}_{unif}[N_i] + \frac{B}{2}\sqrt{B(2\ln 2) \cdot \mathrm{KL}(\mathcal{Q},\mathcal{P})}.^2 \tag{7}$$

According to Lemma 4.4 and our choice of parameters, we have $B(2\ln 2) \cdot \mathrm{KL}(\mathcal{Q},\mathcal{P}) \leq 2B \cdot (\epsilon^2/\sigma^2) \leq 1/4$. Summing both sides of (7) over arm $i$, and using the fact that $\sum_i \mathbb{E}_{unif}[N_i] = B$, we get $\sum_i \mathbb{E}_i[N_i] \leq B + BK/4$, which implies the existence of some $i$ such that $\mathbb{E}_i[N_i] \leq B/K + B/4 \leq (3/4)B$. Therefore, if we choose this distribution $\mathcal{D}_i$, the conditional expected regret of algorithm $\mathcal{A}$ in this interval is at least $\epsilon(B - \mathbb{E}_i[N_i]) \geq \epsilon B/4$.

By choosing distributions inductively in this way, we can make $\mathcal{A}$ suffer a total expected regret of at least $(T/B) \cdot (\epsilon B/4) \geq \Omega(\sqrt[3]{\Lambda V T})$. This completes the proof for the full-information case.

Next, let us consider the bandit case. From Theorem 4.1, we immediately have a lower bound of $\Omega(\sqrt{\Gamma T}) \geq \Omega(\sqrt{VT})$, which implies the required bound when $\sqrt{VT} \geq \sqrt[3]{\Lambda V T}$. When $\sqrt{VT} \leq \sqrt[3]{\Lambda V T}$, we have $V \leq \Lambda^2/T$ which implies that $V \leq \sqrt[3]{\Lambda V T}$, and we can then use the full-information bound of $\Omega(\sqrt[3]{\Lambda V T})$ just proved before. This completes the proof of the theorem. $\square$

## Footnotes

[1]For simplicity of presentation, let us assume here and later in the paper that taking divisions and roots to produce blocks of time steps all yield integers. It is easy to modify our analysis to the general case without affecting the order of our regret bound.

[2]Note that inside the square root, we use $B$ instead of $\mathbb{E}_{unif}[N_i]$ as in [2]. This is because in their bandit setting, $N_i$ is the number of steps when arm $i$ is sampled and has its information revealed to the learner, while in our full-information case, information about arm $i$ is revealed in every step and there are at most $B$ steps.

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
