[Supplementary Material]

# Tracking the Best Expert in Non-stationary Stochastic Environments
# (Supplementary Material)

**Chen-Yu Wei**    **Yi-Te Hong**    **Chi-Jen Lu**
Institute of Information Science
Academia Sinica, Taiwan
{bahh723, ted0504, cjlu}@iis.sinica.edu.tw

## A    Summary and Comparison of Regret Bounds

In Table 1, we list the bounds we derive as well as those from previous works. As can be seen, we have completed a basic picture of the regret landscape characterized by the parameters $\Gamma, V, \Lambda$, and $T$, in both the full-information and bandit settings. Note that we did not provide a bandit upper bound for switching distributions because a good upper bound is already shown in [6]. Although that upper bound does not depend on $\Lambda$ which may lead one to wonder if a better bound is possible with a smaller $\Lambda$, we show that it is in fact impossible by providing a matching lower bound using distributions with $\Lambda = 0$. Let us remark that although an $\Omega(\sqrt{\Gamma T})$ lower bound was also given in [6], it was achieved by distributions with $\Lambda \geq \Omega(T)$. Moreover, the upper bounds of previous works, such as [6, 2, 3], do not take $\Lambda$ into account, so their algorithms do not seem to produce our regret bounds in terms of $\Lambda$. The full-information algorithms of [8, 5] can produce a regret bound of the form $\mathcal{O}(\sqrt{\Gamma L^*})$ for switching distributions, with the quantity $L^*$ being the smallest accumulated loss by an expert sequence that changes at most $\Gamma$ times. Finally, in the full-information setting, [7] provides a regret upper bound which resembles that of ours for drifting distributions but using slightly different definition of parameters.

Table 1: Summary of regret bounds for switching and drifting distributions

| Setting | | This paper | | [8, 5] | | [6] | | [2, 3] | |
|---------|---------------|-----------|-----------|-------|-------|-------|-------|-------|-------|
| Feedback | Distributions | Upper | Lower | Upper | Lower | Upper | Lower | Upper | Lower |
| Full-info | Switching | $\sqrt{\Gamma\Lambda}+\Gamma$ | $\sqrt{\Gamma\Lambda}+\Gamma$ | $\sqrt{\Gamma L^*}$ | - | - | - | - | - |
| Full-info | Drifting | $\sqrt[3]{V\Lambda T}+V$ | $\sqrt[3]{V\Lambda T}+V$ | - | - | - | - | $\sqrt[3]{VT^2}$ | $\sqrt[3]{VT^2}$ |
| Bandit | Switching | - | $\sqrt{\Gamma T}$ | - | - | $\sqrt{\Gamma T}$ | $\sqrt{\Gamma T}$ | - | - |
| Bandit | Drifting | $\sqrt[3]{V\Lambda T}+\sqrt{VT}$ | $\sqrt[3]{V\Lambda T}+\sqrt{VT}$ | - | - | - | - | $\sqrt[3]{VT^2}$ | $\sqrt[3]{VT^2}$ |

## B    Proof of Theorem 3.1

To prove the theorem, we rely on the following key lemma, which we prove in Subsection B.1.

**Lemma B.1.** *Consider a time interval $\mathcal{I}$. Let $V_{\mathcal{I}} = \sum_{t\in\mathcal{I}} \|\mu_t - \mu_{t-1}\|_\infty$ and $\Lambda_{\mathcal{I}} = \sum_{t\in\mathcal{I}} \mathbb{E}\left[\|\ell_t - \mu_t\|_2^2\right]$ be the drift and variance, respectively, in $\mathcal{I}$. Then the expected regret of Algorithm 1 in $\mathcal{I}$ is at most $\tilde{\mathcal{O}}\left(K\sqrt{\Lambda_{\mathcal{I}}} + K + |\mathcal{I}|V_{\mathcal{I}}\right).$*

By applying Lemma B.1 on each interval, we can bound the total regret of Algorithm 1 by

$$\sum_{\gamma=1}^{T/B} \tilde{\mathcal{O}}\left(K\sqrt{\Lambda_{\mathcal{I}_\gamma}} + K + BV_{\mathcal{I}_\gamma}\right) \leq \tilde{\mathcal{O}}\left(K\sqrt{T\Lambda/B} + KT/B + BV\right)$$

from the Cauchy–Schwarz inequality. This gives the bound of $\tilde{O}(\sqrt[3]{K^2\Lambda VT})$ with $B = \sqrt[3]{K^2\Lambda T/V^2}$ when $K\Lambda^2 \geq TV$, while the bound becomes $\tilde{\mathcal{O}}(\sqrt{KVT})$ with $B = \sqrt{KT/V}$ otherwise. Combining these two bounds together gives the regret bound of the theorem.

## B.1 Proof of Lemma B.1

Let us first consider a time step $t$ and bound the regret of our algorithm at that step, which is $\mu_{t,a_t} - \mu_{t,u_t^*}$, where $a_t$ denotes the arm chosen by our algorithm and $u_t^*$ is the best arm. Let $\bar{\mu}_{t,i} = \sum_{s \in S_{t,i}} \frac{\mu_{s,i}}{|S_{t,i}|}$, which is the expected value of $\hat{\mu}_{t,i}$. Note that when arm $a_t$ is pulled, we have $\hat{\mu}_{t,a_t} - \lambda_{t,a_t} \leq \hat{\mu}_{t,u_t^*} - \lambda_{t,u_t^*}$. This implies

$$\bar{\mu}_{t,a_t} - \bar{\mu}_{t,u_t^*} \leq 2\lambda_{t,a_t}$$

when $|\bar{\mu}_{t,i} - \hat{\mu}_{t,i}| \leq \lambda_{t,i}$ for every arm $i$, which happens with probability $1 - \delta K$ by Theorem 2.1 and a union bound. Using the fact that $|\bar{\mu}_{t,i} - \mu_{t,i}| \leq V_{\mathcal{I}}$ for any arm $i$, we have

$$\mu_{t,a_t} - \mu_{t,u_t^*} \leq \bar{\mu}_{t,a_t} - \bar{\mu}_{t,u_t^*} + 2V_{\mathcal{I}} \leq 2\lambda_{t,a_t} + 2V_{\mathcal{I}}$$

with probability $1 - \delta K$.

As a result, by summing over $t \in \mathcal{I}$, the total regret of our algorithm is at most

$$2\sum_{t \in \mathcal{I}} \lambda_{t,a_t} + 2V_{\mathcal{I}}|\mathcal{I}| + \delta K|\mathcal{I}|, \tag{1}$$

with the last term being at most 1 for $\delta \leq 1/(KT)$. It remains to bound the first term above, for which we rely on the following lemma.

**Lemma B.2.** *For any time step $t$ and any arm $i$ such that $|S_{t,i}| > 0$,*

$$\lambda_{t,i} \leq \tilde{\mathcal{O}}\left(\frac{1}{|S_{t,i}|}\left(\sqrt{\sum_{r \in S_{t,i}} \hat{\sigma}_{r,i}^2} + 1\right) + V_{\mathcal{I}}\right), \quad \text{where } \hat{\sigma}_{r,i}^2 = (\ell_{r,i} - \mu_{r,i})^2.$$

Before proving the lemma, let us first use it to bound the first term in (1). Let us divide $\mathcal{I}$ into two parts: $\mathcal{I}' = \{t \in \mathcal{I} : |S_{t,a_t}| \geq 2\}$ and $\mathcal{I} \setminus \mathcal{I}'$. Then $\sum_{t \in \mathcal{I}'} \lambda_{t,a_t}$ is at most

$$\tilde{\mathcal{O}}\left(\sum_{t \in \mathcal{I}'} \frac{1}{|S_{t,a_t}|}\left(\sqrt{\sum_{r \in \mathcal{I}'} \hat{\sigma}_{r,a_r}^2} + 1\right) + |\mathcal{I}'|V_{\mathcal{I}'}\right) \leq \tilde{\mathcal{O}}\left(K\sqrt{\sum_{r \in \mathcal{I}'} \hat{\sigma}_{r,a_r}^2} + K + |\mathcal{I}|V_{\mathcal{I}}\right),$$

since $\sum_{t \in \mathcal{I}'} \frac{1}{|S_{t,a_t}|} \leq K \sum_{t \in \mathcal{I}} \frac{1}{t} \leq K \ln|\mathcal{I}|$, and recall that we use $\tilde{\mathcal{O}}(\cdot)$ to hide logarithmic factors. Then by combining all the bounds together, we have

$$\sum_{t \in \mathcal{I}} \lambda_{t,a_t} \leq 2K + \sum_{t \in \mathcal{I}'} \lambda_{t,a_t} \leq \tilde{\mathcal{O}}\left(K\sqrt{\sum_{t \in \mathcal{I}} \hat{\sigma}_{t,a_t}^2} + K + |\mathcal{I}|V_{\mathcal{I}}\right).$$

Substituting this into the bound in (1) and taking its expectation, we can conclude that the expected regret of our algorithm is at most

$$\tilde{\mathcal{O}}\left(K\sqrt{\sum_{t \in \mathcal{I}} \sum_{i \in [K]} \mathbb{E}\left[\hat{\sigma}_{r,i}^2\right]} + K + |\mathcal{I}|V_{\mathcal{I}}\right) = \tilde{\mathcal{O}}\left(K\sqrt{\Lambda_{\mathcal{I}}} + K + |\mathcal{I}|V_{\mathcal{I}}\right),$$

by Jensen's inequality and the definition of $\Lambda_{\mathcal{I}}$. This gives the regret bound claimed by Lemma B.1. Thus, to complete the proof, it remains to prove Lemma B.2, which we do next.

*Proof of Lemma B.2.* Consider any $t$ and $i$, and for ease of presentation, let us drop the indices involving $t$ and $i$. To bound $\lambda = \rho(|S|, \hat{\Lambda}, \delta)$, defined in Theorem 2.1, let us first bound $\hat{\Lambda} = \sum_{r,s \in S} (\ell_r - \ell_s)^2 / (|S|(|S| - 1))$. Note that each $(\ell_r - \ell_s)^2$ can be expressed as

$$((\ell_r - \mu_r) - (\ell_s - \mu_s) + (\mu_r - \mu_s))^2 \leq 3\left((\ell_r - \mu_r)^2 + (\ell_s - \mu_s)^2 + (\mu_r - \mu_s)^2\right) \quad (2)$$
$$\leq 3\left(\hat{\sigma}_r^2 + \hat{\sigma}_s^2 + V^2\right),$$

using the Cauchy–Schwarz inequality as well as the definition $(\ell_r - \mu_r)^2 = \hat{\sigma}_r^2$ and the fact $(\mu_r - \mu_s)^2 \leq V^2$. Thus, we have

$$\hat{\Lambda} = \sum_{r,s \in S} \frac{(\ell_r - \ell_s)^2}{|S|(|S| - 1)} \leq \mathcal{O}\left(\sum_{r \in S} \frac{\hat{\sigma}_r^2}{|S|} + V^2\right).$$

By plugging this bound into the definition of $\lambda$ and using $\tilde{\mathcal{O}}(\cdot)$ to hide logarithmic factors, we have

$$\lambda \leq \tilde{\mathcal{O}}\left(\sqrt{\sum_{r \in S} \frac{\hat{\sigma}_r^2}{|S|^2} + V^2} + \frac{1}{|S|}\right) \leq \tilde{\mathcal{O}}\left(\frac{1}{|S|}\left(\sqrt{\sum_{r \in S} \hat{\sigma}_r^2} + 1\right) + V\right),$$

since $\sqrt{a + b} \leq \sqrt{a} + \sqrt{b}$ for any $a, b \geq 0$. This proves the lemma. $\qquad\square$

## C  Proof of Theorem 3.2

Recall that our Algorithm 2 is based on that of [4] which is known to have a good regret bound against any offline comparator which does not switch arms. As we consider the dynamic regret here, we need to extend the bound to work against offline comparators which can switch arms. The following lemma provides such a bound.

**Lemma C.1.** *With the choice of $\eta = \sqrt{N/(\Lambda + KV)}$, the regret of Algorithm 2, against an offline comparator switching arms less than $N$ times, is at most $\mathcal{O}(\sqrt{N(\Lambda + KV)})$.*

*Proof.* Consider any offline comparator which switches arms $N - 1$ times, say at $t_1, \ldots, t_{N-1}$. More specifically, assuming $t_0 = 1$ and $t_N = T + 1$, the arm it plays at step $t \in [t_n, t_{n+1})$ remains the same as that at step $t_n$, for any $0 \leq n \leq N - 1$. Let $\pi_t$ denote the characteristic vector of the arm it plays at step $t$. Then when compared against it, the expected regret of our algorithm at step $t$ with respect to a realized loss vector $\ell_t$ (sampled from the distribution) is

$$\sum_{i=1}^{K} \ell_{t,i} \left(\hat{x}_{t,i} - \pi_{t,i}\right) = \langle \ell_t, \hat{x}_t - \pi_t \rangle,$$

which according to [4] is at most

$$\eta \|\ell_t - \ell_{t-1}\|_2^2 + \frac{1}{2\eta}\left(\|\pi_t - \hat{x}_t\|_2^2 - \|\pi_t - \hat{x}_{t+1}\|_2^2\right).^1$$

To bound the total expected regret with respect to these realized loss vectors, let us sum the above over $t$ and observe that the second term above can telescope within each interval and have only $N$ terms remaining at the boundaries between intervals. More precisely, we have

$$\sum_{t=1}^{T}\left(\|\pi_t - \hat{x}_t\|_2^2 - \|\pi_t - \hat{x}_{t+1}\|_2^2\right) = \sum_{n=0}^{N-1}\sum_{t=t_n}^{t_{n+1}-1}\left(\|\pi_{t_n} - \hat{x}_t\|_2^2 - \|\pi_{t_n} - \hat{x}_{t+1}\|_2^2\right)$$
$$\leq \sum_{n=0}^{N-1}\|\pi_{t_n} - \hat{x}_{t_n}\|_2^2,$$

which is at most $2N$ as $\|\pi_n - \hat{x}_{t_n}\|_2^2 \leq 2$. Finally, by taking an additional expectation over the randomness of these loss vectors, the resulting expected regret of Algorithm 2 can be bounded from above by

$$\eta \sum_{t=1}^{T} \mathbb{E}\left[\|\ell_t - \ell_{t-1}\|_2^2\right] + \frac{N}{\eta}.$$

To bound the first sum above, note that using the notation $\sigma_\tau^2 = \mathbb{E}\left[\|\ell_\tau\|_2^2\right] - \|\mu_\tau\|_2^2$, we have

$$\begin{aligned}
\mathbb{E}\left[\|\ell_t - \ell_{t-1}\|_2^2\right] &= \mathbb{E}\left[\|\ell_t\|_2^2\right] + \mathbb{E}\left[\|\ell_{t-1}\|_2^2\right] - 2\langle \mu_t, \mu_{t-1}\rangle \\
&= \sigma_t^2 + \sigma_{t-1}^2 + \|\mu_t\|_2^2 + \|\mu_{t-1}\|_2^2 - 2\langle \mu_t, \mu_{t-1}\rangle \\
&= \sigma_t^2 + \sigma_{t-1}^2 + \|\mu_t - \mu_{t-1}\|_2^2.
\end{aligned} \tag{3}$$

This implies that the expected regret of Algorithm 2 is at most

$$2\eta \sum_{t=0}^{T} \sigma_t^2 + \eta \sum_{t=1}^{T} \|\mu_t - \mu_{t-1}\|_2^2 + \frac{N}{\eta} \leq 2\eta\Lambda + \eta KV + \frac{N}{\eta},$$

as $\sigma_t^2 = \mathbb{E}\left[\|\ell_t - \mu_t\|_2^2\right]$ and $\sum_{t=0}^{T} \mathbb{E}\left[\|\ell_t - \mu_t\|_2^2\right] \leq \Lambda$, using the convention that $\sigma_0 = 0$. Then the lemma follows by choosing $\eta = \sqrt{N/(\Lambda + KV)}$. $\qquad\square$

For switching distributions parameterized by $\Gamma$, we have $V \leq \Gamma$, and by choosing $N = \Gamma$ and $\eta = \sqrt{\Gamma/(\Lambda + K\Gamma)}$, Lemma C.1 provides a regret bound of $\mathcal{O}(\sqrt{\Gamma(\Lambda + K\Gamma)}) \leq \mathcal{O}(\sqrt{\Gamma\Lambda} + \sqrt{K}\Gamma)$ for Algorithm 2.

## D   Proof of Theorem 3.3

Let us first focus on a single interval $\mathcal{I} = [t_1, t_2]$ of time steps, and consider the simpler case of static regret, measured against a single fixed arm. Recall that our learning rates are set as

$$\eta_t = 1/\sqrt{4\sum_{\tau=t_1}^{t-1} \|\ell_\tau - \ell_{\tau-1}\|_2^2}$$

for $t \in \mathcal{I}$, with $\eta_{t_1} = \infty$ initially for every interval. Note that our learning rates are set similarly to those used in [7]. The main difference is that we allow them to be $\infty$, which prevents us from applying their result directly. Still, similarly to [7, Lemma 1], we can have the following guarantee.

**Lemma D.1.** *Using the learning rates given above, the regret of Algorithm 2 in any interval $\mathcal{I} = [t_1, t_2]$ with respect to any fixed arm is at most $\mathcal{O}\left(\sqrt{\sum_{t\in\mathcal{I}}\|\ell_t - \ell_{t-1}\|_2^2}\right)$.*

*Proof.* Let $\rho$ be the first time step in $\mathcal{I}$ such that $\|\ell_\rho - \ell_{\rho-1}\|_2 \neq 0$, so that for any step $t < \rho$ in $\mathcal{I}$, we have $\eta_t = \infty$ and $\ell_t = \ell_{t-1}$. This implies that for any step $t < \rho$ in $\mathcal{I}$, what Algorithm 2 plays is $\hat{x}_t = \arg\min_{x\in\mathcal{X}}\langle \ell_{t-1}, x\rangle = \arg\min_{x\in\mathcal{X}}\langle \ell_t, x\rangle$, which is optimal and has no positive regret. Therefore, it suffices to bound the regret accumulated from step $\rho$ to step $t_2$, which we can do by applying [7, Lemma 1]. To see this, note that the bound there works for any arbitrary starting point $\hat{x}_\rho$ (here computed according to $\eta_\rho = \infty$), and after that we indeed have $\eta_t < \infty$ for $t > \rho$ in $\mathcal{I}$. Then according to [7, Lemma 1], we can upper-bound the regret accumulated from step $\rho$ to step $t_2$ by

$$\sum_{t=\rho}^{t_2} \frac{\eta_{t+1}}{2}\|\ell_t - \ell_{t-1}\|_2^2 + \frac{8}{\eta_{t_2+1}} = \mathcal{O}\left(\sqrt{\sum_{t=t_1}^{t_2}\|\ell_t - \ell_{t-1}\|_2^2}\right).$$

This proves the lemma. $\qquad\square$

The regret bound above is stated in terms of any specific realization of loss vectors. By taking the expectation over their randomness, we can upper-bound the expected regret by

$$\mathcal{O}\left(\mathbb{E}\left[\sqrt{\sum_{t=t_1}^{t_2}\|\ell_t - \ell_{t-1}\|_2^2}\right]\right) \leq \mathcal{O}\left(\sqrt{\sum_{t=t_1}^{t_2}\mathbb{E}\left[\|\ell_t - \ell_{t-1}\|_2^2\right]}\right),$$

using Jensen's inequality, which according to (3) in the proof of Lemma C.1 is at most

$$\mathcal{O}\left(\sqrt{\sum_{t=t_1-1}^{t_2}\sigma_t^2 + \sum_{t=t_1}^{t_2}\|\mu_t - \mu_{t-1}\|_2^2}\right) \leq \mathcal{O}\left(\sqrt{\sum_{t=t_1-1}^{t_2}\sigma_t^2} + \sqrt{K}\sum_{t=t_1}^{t_2}\|\mu_t - \mu_{t-1}\|_\infty\right),$$

using the fact that $\sqrt{a+b} \leq \sqrt{a} + \sqrt{b}$ for $a, b \geq 0$ and $\|\mu_t - \mu_{t-1}\|_2^2 \leq K\|\mu_t - \mu_{t-1}\|_\infty^2$.

By summing such bounds for the $T/B$ intervals, we can upper bound the total regret by

$$\mathcal{O}\left(\sqrt{(T/B)\sum_{t=0}^{T}\sigma_t^2} + \sqrt{K}\sum_{t=1}^{T}\|\mu_t - \mu_{t-1}\|_\infty\right) \leq \mathcal{O}(\sqrt{(T/B)\Lambda} + \sqrt{K}V)$$

using the Cauchy–Schwarz inequality and the definition of $\Lambda$ and $V$. Note that the regret above works against the best offline comparator which plays a fixed arm in each interval (possibly different arms in different intervals), while our goal is to compete with any fully dynamic comparator which can switch arms at every step. Therefore, we would like to bound the difference between the total losses of these two comparators. This can be done similarly to that in the proof of Lemma B.1. More precisely, let us focus on one interval $\mathcal{I}$, and we claim that the difference between their losses in $\mathcal{I}$ is at most $2BV_\mathcal{I}$, where $V_\mathcal{I} = \sum_{t\in\mathcal{I}}\|\mu_t - \mu_{t-1}\|_\infty$ is the total drift in $\mathcal{I}$. To see this, suppose arm $k$ is the best fixed arm in $\mathcal{I}$, which implies that it must the best arm at some step $s \in \mathcal{I}$. Then for any step $t \in \mathcal{I}$, it has $\mu_{t,k} \leq \mu_{s,k} + V_\mathcal{I}$ while the best arm at step $t$, denoted as $u_t*$, has $\mu_{t,u_t*} \geq \mu_{s,u_t*} - V_\mathcal{I} \geq \mu_{s,k} - V_\mathcal{I}$. Therefore, we have

$$\sum_{t\in\mathcal{I}}(\mu_{t,k} - \mu_{t,u_t*}) \leq \sum_{t\in\mathcal{I}}((\mu_{s,k} + V_\mathcal{I}) - (\mu_{s,k} - V_\mathcal{I})) \leq 2BV_\mathcal{I}.$$

Summing over the intervals, we can then upper-bound their total difference by $2BV$.

As a result, we can conclude that the total regret of our algorithm against any fully dynamic comparator is at most $\mathcal{O}(\sqrt{(T/B)\Lambda} + \sqrt{K}V + BV)$. This bound is at most $\mathcal{O}(\sqrt[3]{\Lambda V T} + \sqrt{K}V)$ by setting $B = \sqrt[3]{\Lambda T/V^2}$ when $\Lambda T > V^2$ and at most $\mathcal{O}(\sqrt{K}V)$ by setting $B = 1$ otherwise. Thus, we have the required bound in each case, which proves the theorem.

## E    Proof of Theorem 3.4

Let $R_{T,k}$ denote the regret of our Optimistic-Adapt-ML-Prod algorithm (Algorithm 3) with respect to expert $k$, which is defined as

$$R_{T,k} = \sum_{t=1}^{T} r_{t,k} = \sum_{t=1}^{T}(\langle p_t, \ell_t\rangle - \ell_{t,k}).$$

First, as in [5, Theorem 3], we can bound the regret $R_{T,k}$ in the following form.

**Lemma E.1.** *For any $k$, $R_{t,k}$ is at most*

$$\frac{1}{\eta_{0,k}}\ln\frac{1}{w_{0,k}} + \sum_{t=1}^{T}\eta_{t-1,k}(r_{t,k} - m_{t,k})^2 + \frac{1}{\eta_{T,k}}\ln\left(1 + \frac{1}{e}\sum_{k'=1}^{K}\sum_{t=1}^{T}\left(\frac{\eta_{t-1,k'}}{\eta_{t,k'}} - 1\right)\right). \quad (4)$$

*Proof.* Following [5] we derive (4) by bounding $\ln W_T$ from above and from below. For the lower bound, one can use a similar inductive proof as in [5] to show that for every expert $k \in [K]$,

$$\ln W_T \geq \ln w_{T,k} = \frac{\eta_{T,k}}{\eta_{0,k}}\ln w_{0,k} + \eta_{T,k}\sum_{t=1}^{T}\left(r_{t,k} - \eta_{t-1,k}(r_{t,k} - m_{t,k})^2\right).$$

For the upper bound, some care is needed as we play $\tilde{w}_t$ instead of $w_t$ at step $t$. As in [5], we would like to upper-bound each $W_{t+1}$ in terms of $W_t$. For this, consider any $k$, and note that

$$w_{t+1,k} \leq (w_{t+1,k})^{\frac{\eta_{t,k}}{\eta_{t+1,k}}} + \frac{1}{e}\left(\frac{\eta_{t,k}}{\eta_{t+1,k}} - 1\right),$$

using the fact that $x \leq x^\alpha + (\alpha - 1)/e$ for any $x > 0$ and $\alpha \geq 1$ (see [5, Lemma 13]). The first term on the right-hand side above can be bounded as

$$
\begin{aligned}
(w_{t+1,k})^{\frac{\eta_{t,k}}{\eta_{t+1,k}}} &= w_{t,k}\exp\left(\eta_{t,k}r_{t+1,k} - \eta_{t,k}^2(r_{t+1,k} - m_{t+1,k})^2\right) \\
&= \tilde{w}_{t,k}\exp\left(\eta_{t,k}(r_{t+1,k} - m_{t+1,k}) - \eta_{t,k}^2(r_{t+1,k} - m_{t+1,k})^2\right) \\
&\leq \tilde{w}_{t,k}(1 + \eta_{t,k}(r_{t+1,k} - m_{t+1,k})),
\end{aligned}
$$

using the fact that $\exp(x - x^2) \leq 1 + x$ for any $x \geq -1/2$. Summing the above over $k$, we obtain

$$
\begin{aligned}
\sum_{k=1}^{K}(w_{t+1,k})^{\frac{\eta_{t,k}}{\eta_{t+1,k}}} &\leq \sum_{k=1}^{K}\tilde{w}_{t,k} + \sum_{k=1}^{K}\tilde{w}_{t,k}\eta_{t,k}r_{t+1,k} - \sum_{k=1}^{K}\tilde{w}_{t,k}\eta_{t,k}m_{t+1,k} \\
&\leq \sum_{k=1}^{K}\tilde{w}_{t,k}\exp(-\eta_{t,k}m_{t+1,k}) + \sum_{k=1}^{K}\tilde{w}_{t,k}\eta_{t,k}r_{t+1,k},
\end{aligned}
$$

using the fact that $1 - x \leq \exp(-x)$ for any $x$. The first sum equals $\sum_{k=1}^{K}w_{t,k} = W_t$ by definition. The second sum above equals to zero because $\tilde{w}_{t,k}\eta_{t,k} \propto p_{t+1,k}$ so that the sum is a multiple of $\sum_k p_{t+1,k}r_{t+1,k} = \sum_k p_{t+1,k}(\langle p_{t+1}, \ell_{t+1}\rangle - \ell_{t+1,k}) = 0$. As a result, we have

$$
W_{t+1} = \sum_{k=1}^{K}w_{t+1,k} \leq W_t + \sum_{k=1}^{K}\frac{1}{e}\left(\frac{\eta_{t,k}}{\eta_{t+1,k}} - 1\right)
$$

for any $t$. This implies that

$$
W_T \leq 1 + \frac{1}{e}\sum_{k=1}^{K}\sum_{t=1}^{T}\left(\frac{\eta_{t-1,k}}{\eta_{t,k}} - 1\right),
$$

as $W_0 = 1$. Connecting the upper bound and the lower bound for $\ln W_T$, we have

$$
\sum_{t=1}^{T}r_{t,k} \leq \frac{1}{\eta_{0,k}}\ln\frac{1}{w_{0,k}} + \sum_{t=1}^{T}\eta_{t-1,k}(r_{t,k} - m_{t,k})^2 + \frac{1}{\eta_{T,k}}\ln\left(1 + \frac{1}{e}\sum_{k=1}^{K}\sum_{t=1}^{T}\left(\frac{\eta_{t-1,k}}{\eta_{t,k}} - 1\right)\right),
$$

which proves the lemma. $\square$

Note that as in [5, Theorem 3], the third term in (4) is the price paid for using adaptive learning rates, while our second term is different, which depends on $r_{t,k} - m_{t,k}$ instead of $r_{t,k}$. Next, similarly to [5, Corollary 4], by using the the prior $w_0 = (1/K, \ldots, 1/K)$ and the specific learning rates defined in Eq. (6), one can show that

$$
R_{T,k} \leq \frac{C_{T,K}}{\sqrt{\ln K}}\sqrt{1 + \left(\sum_{t=1}^{T}(r_{t,k} - m_{t,k})^2\right)} + D_{T,K},
$$

where $C_{T,K} = \ln K + \ln(1 + \frac{K}{e}(1 + \ln(T+1)))$ and $D_{T,K} = \frac{1}{4}(\ln K + \ln(1 + \frac{K}{e}(1 + \ln(T+1)))) + 2\sqrt{\ln K} + 16\ln K$. We omit the proof because it is almost identical to that of [5, Proof of Corollary 4], except we have $r_{t,k} - m_{t,k}$ in place of $r_{t,k}$ and we rely on the condition $|r_{t,k} - m_{t,k}| \leq 2$ instead of $|r_{t,k}| \leq 1$. Using the notation $\hat{\mathcal{O}}(\cdot)$ to hide the $\ln\ln T$ factor, we obtain

$$
R_{T,k} \leq \hat{\mathcal{O}}\left(\sqrt{\sum_{t=1}^{T}(r_{t,k} - m_{t,k})^2 \ln K} + \ln K\right). \tag{5}
$$

Finally, since

$$
r_{t,k} - m_{t,k} = (\langle p_t, \ell_t\rangle - \ell_{t,k}) - (\langle p_t, \ell_{t-1}\rangle - \ell_{t-1,k}) = \langle p_t, \ell_t - \ell_{t-1}\rangle - (\ell_{t,k} - \ell_{t-1,k}),
$$

we have $(r_{t,k} - m_{t,k})^2 \leq (2\|\ell_t - \ell_{t-1}\|_\infty)^2$. Substituting this into the bound in (5) completes the proof of the theorem.

**Algorithm 1** Modified Optimistic-Adapt-ML-Prod

---
**for** $t = 1, 2, \ldots, T$ **do**

    Call Algorithm 3 on $\tilde{K}$ sleeping experts to obtain its distribution $\tilde{p}_t$ at step $t$.

    Play the distribution $p_t$, with $p_{t,k} \propto \sum_{s=1}^{t} \tilde{p}_{t,(s,k)}$, and then receive loss vector $\ell_t$.

    Compute $\tilde{\ell}_{t,(s,k)} = \mathbb{1}_{s>t} \cdot \langle p_t, \ell_t \rangle + \mathbb{1}_{s \leq t} \cdot \ell_{t,k}$ and $\tilde{m}_{t,(s,k)} = \mathbb{1}_{s \leq t} \cdot m_{t,k}$, for all $s, k$.

    Feed $\tilde{\ell}_t$ as the loss vector and $\tilde{m}_t$ as the estimate vector to Algorithm 3 for update.

**end for**

---

# F    Proof of Theorem 3.5

The new algorithm is shown in Algorithm 1, which runs Algorithm 3 on $\tilde{K} = KT$ sleeping experts. Each expert is indexed by a pair $(s, k)$, with $s \in [1, T]$ and $k \in [K]$, which is asleep before step $s$ and is awake for steps $t \geq s$. Our Algorithm 1 at step $t$ first calls Algorithm 3 to obtain the distribution $\tilde{p}_t$ over $\tilde{K} = KT$ sleeping experts, from which it computes the distribution $p_t$ over $K$ arms, with

$$p_{t,k} = \sum_{s=1}^{t} \tilde{p}_{t,(s,k)}/Z_t, \quad \text{where } Z_t = \sum_{k'=1}^{K} \sum_{s'=1}^{t} \tilde{p}_{t,(s',k')}.$$

Then it plays $p_t$ at step $t$, receives the loss vector $\ell_t$, and suffers the expected loss $\langle p_t, \ell_t \rangle$. Finally, before proceeding to the next step, it updates Algorithm 3 using the modified loss vector $\tilde{\ell}_t$, where

$$\tilde{\ell}_{t,(s,k)} = \begin{cases} \ell_{t,k} & \text{if expert } (s,k) \text{ is awake (when } s \leq t), \\ \langle p_t, \ell_t \rangle & \text{otherwise,} \end{cases}$$

as well as the estimate vector $\tilde{m}_t$, where

$$\tilde{m}_{t,(s,k)} = \begin{cases} m_{t,k} & \text{if expert } (s,k) \text{ is awake (when } s \leq t), \\ 0 & \text{otherwise.} \end{cases}$$

With such a definition, we can relate the instantaneous regret $\tilde{r}_{t,(s,k)}$, defined as $\langle \tilde{p}_t, \tilde{\ell}_t \rangle - \tilde{\ell}_{t,(s,k)}$, of Algorithm 3 to $r_{t,k}$ of Algorithm 1, defined as $\langle p_t, \ell_t \rangle - \ell_{t,k}$, as shown in the following.

**Lemma F.1.** *For any $s, t$ and any $k$, $\tilde{r}_{t,(s,k)} = 0$ if $s > t$ and $\tilde{r}_{t,(s,k)} = r_{t,k}$ otherwise.*

We prove the lemma in Subsection F.1. From this, together with Theorem 3.4, we can bound the regret of Algorithm 1 during an interval against some arm in the following, which we prove in Subsection F.2.

**Lemma F.2.** *For any $t_1 \leq t_2$ and any $k$, $\sum_{t=t_1}^{t_2} r_{t,k} \leq \tilde{\mathcal{O}}\left( \sqrt{\sum_{t=t_1}^{t_2} \|\ell_t - \ell_{t-1}\|_\infty^2 \ln K} + \ln K \right)$.*

Using this lemma, we can show the following dynamic regret bound for our Algorithm 1.

**Lemma F.3.** *For distributions with parameters $\Lambda$ and $V$, the expected regret of Algorithm 1, against any offline comparator switching arms less than $N$ times, is at most $\tilde{\mathcal{O}}\left( \sqrt{N\Lambda \ln K} + V\sqrt{\ln K} + N \ln K \right)$.*

We will prove the lemma in Subsection F.3. With this lemma, we are now ready to prove Theorem 3.5. First, for switching distributions parameterized by $\Gamma$ and $\Lambda$, we can simply choose $N = \Gamma$ in Lemma F.3 and have the regret bound of $\tilde{\mathcal{O}}(\sqrt{\Gamma \Lambda \ln K} + \Gamma \ln K)$ as $V \leq \Gamma$.

Next, for drifting distributions parameterized by $V$ and $\Lambda$, we need to choose $N$ appropriately. As shown in the proof of Theorem 3.3, a regret bound with respect to comparators making less than $N$ equally-spaced switches immediately implies a regret bound with respect to fully-dynamic comparators with an additional term of $\mathcal{O}(VT/N)$. This leads to a dynamic regret bound of $\tilde{\mathcal{O}}(\sqrt{N\Lambda \ln K} + V\sqrt{\ln K} + N \ln K) + \mathcal{O}(VT/N)$ for our algorithm. When $\Lambda^2 \geq VT \ln K$, we can choose $N = \sqrt[3]{V^2 T^2/(\Lambda \ln K)}$ to give a regret bound of $\tilde{\mathcal{O}}(\sqrt[3]{\Lambda V T \ln K})$. On the other hand, when $\Lambda^2 < VT \ln K$, we can choose $N = \sqrt{VT/(\ln K)}$ to give a regret bound of $\tilde{\mathcal{O}}(\sqrt{VT \ln K})$. By combining these two bounds together, we have the theorem.

## F.1 Proof of Lemma F.1

First, we claim that $\langle \tilde{p}_t, \tilde{\ell}_t \rangle = \langle p_t, \ell_t \rangle$ for any $t$, which means that the expected loss of Algorithm 1 equals that of Algorithm 3 at any step. To see this, note that from the definition of $\tilde{\ell}_t$, we have

$$\langle \tilde{p}_t, \tilde{\ell}_t \rangle = \sum_{k=1}^{K} \sum_{s=1}^{t} \tilde{p}_{t,(s,k)} \ell_{t,k} + \sum_{k=1}^{K} \sum_{s=t+1}^{T} \tilde{p}_{t,(s,k)} \langle p_t, \ell_t \rangle,$$

where the first sum above, using the definition of $p_{t,k}$, equals

$$\sum_{k=1}^{K} Z_t p_{t,k} \ell_{t,k} = Z_t \langle p_t, \ell_t \rangle = \sum_{k=1}^{K} \sum_{s=1}^{t} \tilde{p}_{t,(s,k)} \langle p_t, \ell_t \rangle.$$

This implies that

$$\langle \tilde{p}_t, \tilde{\ell}_t \rangle = \sum_{k=1}^{K} \sum_{s=1}^{T} \tilde{p}_{t,(s,k)} \langle p_t, \ell_t \rangle = \langle p_t, \ell_t \rangle.$$

Then for $s > t$, we have $\tilde{r}_{t,(s,k)} = \langle \tilde{p}_t, \tilde{\ell}_t \rangle - \langle p_t, \ell_t \rangle = 0$, while for $s \leq t$, we have $\tilde{r}_{t,(s,k)} = \langle \tilde{p}_t, \tilde{\ell}_t \rangle - \ell_{t,k} = r_{t,k}$, which proves the lemma.

## F.2 Proof of Lemma F.2

According to Lemma F.1 and Theorem 3.4, we have

$$\sum_{t=t_1}^{t_2} r_{t,k} = \sum_{t=t_1}^{t_2} \tilde{r}_{t,(t_1,k)} = \sum_{t=1}^{t_2} \tilde{r}_{t,(t_1,k)} \leq \hat{\mathcal{O}} \left( \sqrt{\sum_{t=1}^{t_2} \left( \tilde{r}_{t,(t_1,k)} - \tilde{m}_{t,(t_1,k)} \right)^2 \ln \tilde{K}} + \ln \tilde{K} \right).$$

Then using Lemma F.1 as well as the definition of $\tilde{m}_{t,(t_1,k)}$, we know that $\tilde{r}_{t,(t_1,k)} - \tilde{m}_{t,(t_1,k)}$ equals $0$ if $t < t_1$ and it equals $r_{t,k} - m_{t,k}$ otherwise. As $(r_{t,k} - m_{t,k})^2 \leq (2\|\ell_t - \ell_{t-1}\|_\infty)^2$ and as $\tilde{K} = KT$, we have

$$\sum_{t=t_1}^{t_2} r_{t,k} = \tilde{\mathcal{O}} \left( \sqrt{\sum_{t=t_1}^{t_2} \|\ell_t - \ell_{t-1}\|_\infty^2 \ln K} + \ln K \right),$$

with the notation $\tilde{\mathcal{O}}(\cdot)$ hiding an additional $\log T$ factor. This proves the lemma.

## F.3 Proof of Lemma F.3

Consider any offline algorithm, which divides the time steps into $N$ intervals: $\mathcal{I}_1, \ldots, \mathcal{I}_N$, and plays the best fixed arm in each interval. For any one such interval $\mathcal{I} = [t_1, t_2]$ and any arm $k$, we know from Lemma F.2 that

$$\sum_{t=t_1}^{t_2} r_{t,k} \leq \tilde{\mathcal{O}} \left( \sqrt{\sum_{t=t_1}^{t_2} \|\ell_t - \ell_{t-1}\|_\infty^2 \ln K} + \ln K \right). \tag{6}$$

From Eq. (2) in the proof of Lemma B.2, we know that

$$\|\ell_t - \ell_{t-1}\|_\infty^2 \leq 3 \left( \|\ell_t - \mu_t\|_\infty^2 + \|\mu_t - \mu_{t-1}\|_\infty^2 + \|\ell_{t-1} - \mu_{t-1}\|_\infty^2 \right).$$

Using the fact that $\|\ell_\tau - \mu_\tau\|_\infty^2 \leq \|\ell_\tau - \mu_\tau\|_2^2$ and $\sqrt{a+b} \leq \sqrt{a} + \sqrt{b}$ for $a, b \geq 0$, we have

$$\sqrt{\sum_{t=t_1}^{t_2} \|\ell_t - \ell_{t-1}\|_\infty^2 \ln K} \leq \sqrt{6 \sum_{t=t_1-1}^{t_2} \|\ell_t - \mu_t\|_2^2 \ln K} + \sum_{t=t_1}^{t_2} \|\mu_t - \mu_{t-1}\|_\infty \sqrt{3 \ln K}.$$

Then taking the expectation on both sides of (6) and applying Jensen's inequality, we have

$$\sum_{t=t_1}^{t_2} \mathbb{E}\left[r_{t,k}\right] \leq \tilde{\mathcal{O}} \left( \sqrt{\sum_{t=t_1-1}^{t_2} \mathbb{E}\left[\|\ell_t - \mu_t\|_2^2\right] \ln K} + \sum_{t=t_1}^{t_2} \|\mu_t - \mu_{t-1}\|_\infty \sqrt{\ln K} + \ln K \right).$$

Note that the bound above work for any arm $k$. Therefor, if we let $k_n$ be the best arm in interval $\mathcal{I}_n$ and sum the above over the intervals, we can bound the expected regret against such a dynamic comparator as

$$\sum_{n=1}^{N} \sum_{t \in \mathcal{I}_n} \mathbb{E}\left[r_{t,k_n}\right] \leq \tilde{\mathcal{O}}\left(\sqrt{N \sum_{t \in [T]} \mathbb{E}\left[\|\ell_t - \mu_t\|_2^2\right] \ln K} + \sum_{t \in [T]} \|\mu_t - \mu_{t-1}\|_\infty \sqrt{\ln K} + N \ln K\right),$$

by the Cauchy–Schwarz inequality. Then the lemma follows from the definition of $\Lambda$ and $V$.

## G  Proof of Theorem 4.2

Fix any full-information algorithm $\mathcal{A}$. Let us consider two cases depending on whether or not $\Lambda > \Gamma$. First, for the case with $\Lambda > \Gamma$, we divide the time horizon into $\Gamma$ intervals of equal length. In the first $\Lambda/\Gamma$ time steps of each interval, according to the lower bound proof in [1], there exists a stationary loss distribution of variance 1 for making the regret of any algorithm $\mathcal{A}$ at least $\Omega(\sqrt{\Lambda/\Gamma})$. For the rest of the intervals we use a fixed distribution with no variance. With this choice of distributions, the total variance does not exceed $\Lambda$, and the regret lower bound is $\Gamma \cdot \Omega(\sqrt{\Lambda/\Gamma}) \geq \Omega(\sqrt{\Gamma\Lambda}) \geq \Omega(\sqrt{\Gamma\Lambda} + \Gamma)$.

Next, for the case with $\Gamma \geq \Lambda$, we show a lower bound of $\Omega(\Gamma) \geq \Omega(\sqrt{\Gamma\Lambda} + \Gamma)$. Let $\ell^{(i)}$, for $i \in [K]$, be the loss vector with 0 in dimension $i$ and 1 elsewhere, which can serve as a distribution with mean $\ell^{(i)}$ and variance 0, with arm $i$ being the optimal one. Then one can show the existence of a sequence of $\Gamma$ loss vectors, chosen from these $K$ vectors, which makes the expected regret of $\mathcal{A}$ at least $\Gamma(K-1)/K$ for the first $\Gamma$ steps, by a probabilistic argument. Formally, by choosing one of the loss vectors uniformly and independently for each step, the expected loss of $\mathcal{A}$ is $\Gamma(K-1)/K$, which implies the existence of a fixed sequence of $\Gamma$ loss vectors achieving this bound. On the other hand, the total loss of a fully dynamic offline algorithm is clearly 0, which implies that the expected regret of $\mathcal{A}$ with respect to this fixed sequence of $\Gamma$ loss vectors is at least $\Gamma(K-1)/K = \Omega(\Gamma)$. The last loss vector is then kept for the remaining $T - \Gamma$ steps, which cannot decrease the regret, and the total number of switches is at most $\Gamma - 1$. This completes the proof of the theorem.

## H  Proof of Lemma 4.4

We let $\mathcal{Q}$ and $\mathcal{P}$ both be Bernoulli-type distributions taking values from $\{0, 2\sigma\}$. For $\mathcal{Q}$, we let it take each value with equal probability so that it has mean $\sigma$ and variance $\sigma^2$. For $\mathcal{P}$, we make it have mean $\sigma - \epsilon$, by letting it take the value 0 with probability $\frac{\sigma+\epsilon}{2\sigma}$ and the value $2\sigma$ with probability $\frac{\sigma-\epsilon}{2\sigma}$. Then the variance of $\mathcal{P}$ is $\frac{\sigma+\epsilon}{2\sigma}(\sigma-\epsilon)^2 + \frac{\sigma-\epsilon}{2\sigma}(\sigma+\epsilon)^2 = (\sigma-\epsilon)(\sigma+\epsilon) < \sigma^2$. Thus, the first two conditions are satisfied. For the last condition, note that

$$(\ln 2)\mathrm{KL}(\mathcal{Q}, \mathcal{P}) = \frac{1}{2}\ln\left(\frac{\frac{1}{2}}{\frac{\sigma-\epsilon}{2\sigma}}\right) + \frac{1}{2}\ln\left(\frac{\frac{1}{2}}{\frac{\sigma+\epsilon}{2\sigma}}\right) = \frac{1}{2}\ln\left(\frac{\sigma^2}{\sigma^2-\epsilon^2}\right) = \frac{1}{2}\ln\left(1 + \frac{\epsilon^2}{\sigma^2-\epsilon^2}\right)$$

which, using the fact that $1 + x \leq e^x$ and $\epsilon^2 \leq \sigma^2/2$, is at most $\frac{\epsilon^2}{2(\sigma^2-\epsilon^2)} \leq \frac{\epsilon^2}{\sigma^2}$ as required.

## Footnotes

[1] Although [4] considered the case with identical $\pi_t = \pi$ for every $t$, it is straightforward to see this from their Lemma 5 and their proof of Theorem 8.