[Reviews · NeurIPS 2016]

Reviewer 1

Summary

The paper considers online learning in a non-stationary stochastic environment, and considers the achievable regret as a function of the change in the mean loss vector (measured by the number of switches or the amount of shift) and the cumulative variance of the random losses. Algorithms are given with regret upper bounds, as well as matching lower bounds.

Qualitative Assessment

ADDED AFTER THE REBUTTAL: - Indeed, the proof of Theorem 4.1 is correct, and the constructed environment is oblivious; it was my mistake, sorry about it. - The generalization in the analysis of UCB-V seems quite straightforward, as--if I recall correctly--the applied concentration inequalities do not require that the random variables have the same distribution (neither in your nor in their case). - Comparison to Jadbabaie et al: If in their work one chooses M to be the previous gradient, then their D_T is essentially the same as your Lambda (up to some constants), and the difference is that you take expectation inside the norms in V. While I might be mistaken here, but it seems to me that with a slight modification of their proof you can move the expectation inside. Nevertheless, you have to check this, and describe the connection to related work with care (what follows easily and what not). - You should still argue why the proposed complexity parameters are interesting, e.g., present an example when Lambda is not zero or linear in T. ----------------------------------------- While the paper might have some interesting contributions, it does not consider the existing literature and does not cite the references adequately. This, in fact, makes it hard to asses the novelty of the results, as several of them seem to easily follow from existing work. In fact, the paper of Jadbabaie, Rakhlin, Shahrampour, and Sridharan, whose arxiv version is "sort of" cited in the paper (included in the references but never discussed properly), which was published in AISTATS 2015, provides an algorithm quite similar to Algorithm 2, which achieves similar bounds, but in the stronger adversarial case. The main addition of the paper to admitted existing work (Besbes et al., 2014--please correct the order of the authors in the reference) is to consider the variance of the random losses. As far as I can see, this is achieved by combining the techniques to derive second order bounds and some sort of shifting bounds (even blocking, in the simplest case). This would be interesting if it had not been done before by Jadbabaie et al. (2015). However, given the previous work, a thorough discussion of novelty would be required. Also, the writing of the paper has to be improved dramatically: e.g., not even the UCB-V algorithm of Audibert et al. (2009) is referenced (although used almost identically). Concerning motivation, other than saying that the complexity measures used in the paper appear in the literature, why are they interesting? For example, if there is a clearly suboptimal arm, why does it matter if its mean changes or it has a relatively high variance (assume that even the smallest loss from this arm is bigger than the losses of any other arm)? What are the interesting cases when \Lambda is not constant or linear in time? Also, what is the connection to tracking/shifting bounds (Herbster and Warmuth 1998, 2001)? Note that in the standard online learning literature, tracking and shifting bounds usually correspond to properties of the comparator sequence, not of the loss functions, so this should be clarified to avoid confusion. The switching results should also relate to the large body of work done in information theory on coding piecewise stationary sources (although there usually the log-loss is considered, but the techniques are essentially the same, although instead of the variance the the entropy/entropy-rate of the source is considered). The sleeping expert-type reductions are also available in generic forms, see e.g., Gyorgy et al. (2012) and the references therein. All in all, I see potential in the paper, but substantial improvements are needed before it can be published. References: J-Y. Audibert, R. Munos, and Cs Szepesvari. Exploration-exploitation trade-off using variance estimates in multi-armed bandits. Theoretical Computer Science, 410:1876-1902, 2009. A. Gyorgy, T. Linder, and G. Lugosi, "Efficient Tracking of Large Classes of Experts," IEEE Transactions on Information Theory, vol. 58, pp. 6709-6725, November 2012. Herbster, M. and Warmuth, M.K. (1998) “Tracking the Best Expert”, in the special issue on context sensitivity and concept drift of the Journal of Machine Learning, Vol. 32(2), pp. 151-178, August 1998. Herbster, M. and Warmuth, M.K. (2001) “Tracking the Best Linear Predictor”, Journal of Machine Learning Research, Vol. 1, pp. 281-309, September 2001.

Confidence in this Review

3-Expert (read the paper in detail, know the area, quite certain of my opinion)


Reviewer 2

Summary

In the stationary setting, with the same distribution for all time steps, a good online player can choose the optimal one with a high confidence. But in the non-stationary world, the distribution may drift as time evolves. In this paper, the authors partition the se-quences into \Gamma intervals, with each interval having a station-ary distribution. They also introduce a parameter V to denote the difference between the means of consecutive distributions and use Λ to measures the total statistical variance of the distributions. The paper try to discuss how the different \Gamma ,Λ and V affect the dynamic pseudo-regret, in both full-information and bandit set-tings.

Qualitative Assessment

The work is well motivated and investigates an interesting problem of tracking the dynamic regret of sequential prediction in the non-stationary stochastic environment. This paper was well-written and generally comprehensible to someone without experience in the on-line learning or the prior work dis-cussed here. The contribution seems to be novel and significant. There were a few places where the presentation could be improved, and some of the results were difficult to evaluate. 1. Line 115: what does a_t mean in l_{t,a_t}? 2. The theorem 2.1 appeared with no discussion. Why did you give out theorem 2.1 here? 3. Algorithm1: why does you use the equidistant interval for [(m-1)B+1,mB]? The distribution may have different lengths of interval. Are you sure that the different partitions will not affect your regret bound (from you Note.1 in page4)? 4. Line 147: How to get the formula (2)? 5. Line 162: What is N? 6. Line 174-175: How to get the formula of B? 7. Line 186&190: The symbolic interpretation should be coincident, such as expert k or arm k? 6. Page6,Line 215 : hides a ln ln T factor? What is the difference of Switching distribution and drifting distribution? Just consider \Gamma for switching and V for drifting? 7. Page5: Actually, it is hard to understand from Sec3.3. What does w_{t,k} and m_{t,k} mean? It seems like Exponential weighted average. How can we get the formula (6). The logic of this section is very confusing. 8. Page7-Sec4: They only discuss the actionss of K=2, which I think it is an appeared limitation here. The paper ended in haste. The references are uncomplete. The following two papers are highly related with this paper: “Tracking the Best Expert ”-Mark Herbster, Manfred K. Warmuth and Switching Portfolios Yoram Singer 98. The author can compare your model and result with them.

Confidence in this Review

2-Confident (read it all; understood it all reasonably well)


Reviewer 3

Summary

The paper considers the problem of online learning with K actions under both full and bandit information, and studies the notion of dynamic regret (regret measured against the strongest comparator that can switch actions in each round). The main novelty in the paper is studying the impact of the total variance of the loss vectors on the dynamic regret: the authors show several new upper and lower bounds in terms of this quantity.

Qualitative Assessment

The paper is well-written and the results are, to my knowledge, novel. While many of the technical tools used in the proofs are more-or-less standard, their combination offers some non-trivial challenges and the execution of the analysis appears to be excellent. My only complaint is that the results are of a somewhat limited interest: none of the proposed algorithms is very practical and the bounds don't give any drastically new insight about the complexity of learning in the considered settings. Besides, I also have a feeling that I'm missing something about the results. Namely, it appears that the newly introduced quantity \Lambda may scale linearly with the number of arms/experts K, so the new bounds may have a worse scaling with this parameter than existing bounds (K^{1/3} instead of log K in the full-information setting). At first glance, the lower bounds of Theorem 4.3 appear to contradict known upper bounds in these terms, but I believe that this is not the case: the problem seems to be that the statement of the theorem is phrased a bit poorly. It should be made clear that the statement of the theorem does *not* hold in a problem-dependent sense, but only in the sense that "for any *fixed* choice of \Lambda, T and V, there exist a sequence on which any algorithm is going to suffer a regret of \Omega(...)". It would be nice if the authors added a discussion about this issue, as it may be puzzling for other readers too. Overall, while I think that the results are certainly not ground-breaking, I don't have problems with the paper being published. Detailed comments ================= 050: Describing the notion of "dynamic regret" after stating the bounds is a bit clumsy; consider rewriting this section so that it begins with the regret definition. 061: Same as above: informal definition of dynamic regret repeated. 089: Technically, your bound does not recover the previous results, as you have a worse scaling in V (sqrt(V) vs. V^{1/3}). To resolve this apparent conflict, you should mention that the asymptotics of V are tied together with the asymptotics of T. 111: "choose an action a_t and then suffer a corresponding loss \ell_{t,i}"---maybe \ell_{t,a_t}? 146: Add reference for vUCB (Audibert, Munos and Szepesvari, 2009?). Theorem 3.1: Since you consider both full-information and bandit settings, it would be useful if the theorems explicitly stated which setting they are about. Eq. (5): Note that this doesn't exactly match the algorithm of [4]; it is more closely related to the Bernstein Online Aggregation algorithm of Wintenberger (arxiv 2014) or to the Variation MW algorithm of Hazan and Kale (COLT 2008). Theorem 3.4: Note that it is possible to get better bounds by using the algorithm of Steinhardt and Liang [10], which depend on the path length of the best expert rather than the L1 path length of the loss vectors. (Also note however that this algorithm is not parameter-free.)

Confidence in this Review

3-Expert (read the paper in detail, know the area, quite certain of my opinion)


Reviewer 4

Summary

The paper considers a sequential decision problem with stochastic rewards bounded in [0,1], both in the full information and bandit information settings. The mean reward of each arm is allowed to change with time, and thus the instantaneous regret at time t is defined with respect to the best action at this time. The cumulative regret is thus defined against a strategy that potentially selects a different arm at each time step. The rewards are assumed to be independent across arms and time. Three quantities are considered that capture the complexity of the environment: the number of times the mean reward vector changes (Gamma-1) the cumulative drift measured in infinite norm (V), and the cumulative variance of the reward distributions (Lambda). In this non-stationary setting, the paper studies several algorithms. Algorithm 1 has perfect knowledge of Gamma, Lambda, V and of the time horizon and works in the bandit setting. Algorithm 2 has similar knowledge and works on the full-information setting. Algorithm 3 is parameter free, and works in the full information setting. It is directly inspired from the work of [3] and [4]. Its regret is controlled in Theorem 3.5; its static regret against any fixed arm is controlled in Theorem 3.4. Section 4 provides lower bounds on the regret, in terms of the quantities Gamma, Lambda, V.

Qualitative Assessment

I have read the feedback from the authors. Overall, this is an interesting contribution. Algorithms 1 and 2 are arguably not interesting; at least it provides some warming-up. Algorithm 3 is more interesting. It uses a simple, yet effective combination of two ideas coming from [3] and [4]. The regret performance is also appealing, especially in view of the lower bounds. However, it could be useful to provide some explicit constants in the main material of the paper. I disagree with the claim that the regret is "constant" with respect to time, since the quantity Lambda typically scales with T, even when Gamma and V are constants. Thus, I think this claim should be nuanced. Any idea how to extend this work to the parameter-free bandit setting? Gamma is defined in terms of the number of changes between the mean vectors. Another related quantity of interest is the number of changes of the optimal arm, which can be arbitrarily smaller than Gamma. Do you think such a quantity could be used as well or do you see a strong reason why not do use it?

Confidence in this Review

3-Expert (read the paper in detail, know the area, quite certain of my opinion)


Reviewer 5

Summary

The paper provides improved lower and upper bounds on the regret in the bandit and the full information setting. The bounds are formulated in terms of four parameters: * the number \Gamma the index of the optimal arm changes, * the sum V of the changes of the mean reward between subsequent rounds, * the sum \Lambda of the variances of the rewards in the various rounds, which has not been considered before in this context, * and the time horizon T. Previous results only combined T with \Gamma or V, and can be obtained from the results in this paper as a kind of worst case version.

Qualitative Assessment

The paper is well-motivated, easy to read, and all the details are clearly described. Although the derived results are not likely the ultimate version of the regret bounds (meaning that the parameters do not capture all the seemingly important features of the problem complexity), they do provide interesting additions to existing results, and clearly contribute to the deeper understanding of the topic. A downside is that most of the algorithms require some knowledge of the parameters which is rather unrealistic. Nevertheless, the paper provides enough additional novelties that make it a valuable contribution. - Line 192: adding the remark "m_t is determined below" could slightly improve the reading experience. - Lines 186-187: do you choose a random arm generated by using distribution p_t? - Line 205: is \eta_{t,k} correct there? Shouldn't it be \eta_{t-1,k}? - Lines 304-309: i is already preserved for the index of the optimal arm. Please use a different index here. - Lines 318-320: Does not \sqrt{VT} \el \sqrt[3]{\Lambda VT}, plugged into Theorem 4.1 immediately imply the claim of the theorem? Is the argument in the sentence between lines 318 and 320 really needed? - Table 1 in Appendix A is very nice and really helpful. However, extending it with highlighting the upper bounds that are based on algorithm that require extra knowledge of the parameters could make it even more helpful. - An overall question: the regret in this paper is computed against the best dynamic learner. What could be the regret against the best fixed arm? That is, could these parameters lead to some improvement of the classical results too?

Confidence in this Review

2-Confident (read it all; understood it all reasonably well)


Reviewer 6

Summary

The paper investigates on the regret of sequential prediction in nonstationary environments. A parameter Λ has been introduced, denoting the variance of the loss distributions over the process. The paper presents a study on such parameter and how it affects the prediction regret. In particular, the interaction of Λ with other distribution measures, such as the number of times distributions change or distribution deviation over time, is investigated. A derivation of the regret lower bound is also shown.

Qualitative Assessment

The paper looks promising but at the same time it seems to be incomplete. Experimental results proving the effectiveness of the proposed algorithms would give robustness and reliability to achieved theoretical results. Abstract is unclear and must be rewritten. Symbols and parameters are not well defined and it is not clear in which context (equations, algorithms,...) they are introduced. The same occurs in the introduction. In page 2, line 44, UCB is not defined. Overall, clearness must be improved. Bibliography must be improved as several works exist in the literature on best expert tracking.

Confidence in this Review

2-Confident (read it all; understood it all reasonably well)